# DEFENDER OF PRIVACY AND FAIRNESS: TINY BUT REVERSIBLE GENERATIVE MODEL VIA MUTUALLY COLLABORATIVE KNOWLEDGE DISTILLATION

## ABSTRACT

Sharing vast amounts of data to train powerful artificial intelligence (AI) models raises public interest concerns such as privacy and fairness. While reversible anonymization techniques are very effective for privacy preservation and fairness enhancement, these methods rely on heavy reversible generative models, making them only suitable to run in the cloud or on a server independent from the image source. For example, data transmission might be under the privacy threats such as channel eavesdropping. Therefore, we propose a novel mutually collaborative knowledge distillation strategy to train a tiny and reversible generative model. This enables us to build a synthesis-based privacy and fairness protection system in embedded devices for anonymizing privacy-sensitive data and thus improve security protection capabilities from the source. The proposed mutually collaborative knowledge distillation method exploits the reversibility of the generative model. By pairing the teacher encoder (decoder) with the student decoder (encoder), we train the student decoder (encoder) by reconstructing the image space (latent space) from the prior image space (latent space). This results in tiny-size student models that can be embedded into devices. We deploy and evaluate our system on NVIDIA Jetson TX2 devices, which operate in real-time. Extensive experiments demonstrate that our system effectively anonymizes face images and thus protects privacy and also improves fairness while minimizing the impact on downstream tasks. Our code will be publicly available on GitHub.

## 1 INTRODUCTION

The impact of sharing large volumes of data for training advanced artificial intelligence (AI) models has been profound, transforming various aspects of human life. However, this progress has also brought forth concerns regarding important societal values like fairness and privacy (Ribaric et al., 2016; Elkoumy et al., 2021). Instances of significant privacy breaches, such as the iCloud leakage event (Wang et al., 2018b) and the ClearView AI incident (Marks, 2021), have prompted the establishment of privacy regulations like the California Consumer Privacy Act (Pardau, 2018) in the United States. Notably, Italy recently became the first Western country to ban ChatGPT due to its perceived threats to human privacy, subsequently inspiring the European Union to introduce the AI ACT (Helberger & Diakopoulos, 2023) aimed at regulating AI ethics and morality. In answering regulations such as General Data Protection Regulation (GDPR) (Regulation, 2018), Trustworthy AI (AI, 2019) and the Social Principles of Human-Centric AI (Secretariat et al., 2019), this paper seeks "privacy by design" principle (Article 25 of the GDPR, that is, integrate data protection from the design stage right through the lifecycle) and "pseudonymization" (Article 32 of the GDPR, that is, effectively pseudonymize/anonymize the face data by replacing the original face image with a fake face image and saving a reversible key) to enhance visual privacy and fairness.

Among the aspects of visual privacy, preserving face identity is particularly crucial (Acquisti et al., 2014; Meden et al., 2021). The simplest anonymization method for face privacy protection is the obfuscation-based approach, which involves blurring, masking, or pixelating the face region to conceal facial information (McPherson et al., 2016), however, leads to permanent loss of information in the obscured image regions, which may not be ideal for downstream tasks (Ren et al., 2018; Gu et al., 2020). To address these limitations, synthesis-based methods have emerged as a popular alternative for face anonymization/pseudonymization to hide the privacy (Mosaddegh et al., 2015; Ren et al., 2018; Zhu et al., 2020; Gu et al., 2020) (synthesis-based methods are more precisely called

pseudonymization as they replace the original face with a fake one, while we follow the more general parlance of calling it anonymization throughput this work). In particular, it replaces original faces with new faces obtained from a pre-existing database or generated fake faces (e.g., human-like faces) superimposed on the original face. State-of-the-art synthesis-based approaches are typically designed based on three key properties: privacy preservation, reversibility, and data availability for downstream tasks. They come with costs. Compared to simple blurring, existing synthesis-based methods require substantial computational resources due to heavy generators, such as generative adversarial networks (GANs) (Goodfellow et al., 2014) or reversible flow-based generative models (Dinh et al., 2014). Consequently, most of these methods are deployed in the cloud or on servers, posing a potential risk of privacy leakage during data transmission from the image sources (Ren et al., 2018; Gu et al., 2020). A recent attempt by Zhu et al. (Zhu et al., 2023) aims to mitigate this risk by embedding part of the privacy protection system in the camera and anonymizing private information with an embedded key. However, due to the large size of the flow-based generative model (Baranchuk et al., 2021), (Zhu et al., 2023) still need to generate anonymized fake faces in the cloud, leaving it vulnerable to internal cloud model attackers (Ye et al., 2022). This somehow violates the principle of "privacy by design" at the early stage in the lifecycle.

This paper aims to address the challenge of embedding a synthesis-based privacy protection system with reversible generators into embedded devices like surveillance cameras or mobile terminals to enhance data security. The key challenge lies in reducing the size of the reversible generator model, and to tackle this, we propose a novel knowledge distillation (KD) strategy. In the literature, existing KD methods are mostly designed for high-level tasks such as image classification (Romero et al., 2014; Ahn et al., 2019), object detection (Chen et al., 2017; Dai et al., 2021), and semantic segmentation (Liu et al., 2019; Qin et al., 2021), while others focus on low-level tasks like image generation (Chen et al., 2020; Li et al., 2020) and style transfer (Wang et al., 2020). In this work, we bridge the gap by proposing a mutually collaborative knowledge distillation strategy specifically tailored for generators with encoder-decoder architectures. This strategy is inspired by (Wang et al., 2020) where a collaborative KD method is proposed for style transfer. Different from collaborative KD which only distills the decoder from the encoder, the proposed KD process mutually distills the student encoder (decoder) from the teacher decoder (encoder), which ensures the reversibility between the encoder and decoder. It is worth noting that reversibility ensures privacy-sensitive data can be restored if needed, which is important in the anonymization/pseudonymization process.

In particular, we choose the flow-based generator (Zhu et al., 2023) as the teacher model which exhibits inherent reversibility property. For the generators with an encoder-decoder pair, traditional KD usually transfers knowledge from the teacher encoder (decoder) to the student encoder (decoder) by minimizing a loss function defined on the output space of the teacher model as shown in Figure 1(a). The teacher encoder may surrogate some performance degradation due to its training process, which may induce a distorted output space and degrade the performance of the student encoder. This performance degradation also happened to the student decoder. In Figure 1(b), the reversibility of the encoder-decoder pair enables the student decoder to reconstruct the image space from the input of the teacher encoder. Hence, we can distill the student decoder from the teacher encoder by minimizing the distance between the reconstructed image space and the prior image space; and distill the student encoder from the teacher decoder by minimizing the distance between the reconstructed latent space and the prior latent space. Armed by this mutually collaborative KD method, the compressed student encoder and decoder are both tiny-size and thus can be put into the embedded system, thereby replacing the original face with a generated fake face to protect face privacy. As shown in Figure 2. with different de-anonymization orders, our embedded protection system can support users with different privacy needs. For example, it sends the face-aimed users the original face; sends obfuscation-tolerant users the anonymized latent-variance face (ALF) which is obtained by putting the original face into the student encoder and then rotating by an anonymization key; sends the privacy-aimed users the fake face (FF) which is obtained by putting the original face into the student encoder, rotating by an anonymization key and passing through the student decoder. The model reversibility ensures that the original face can be restored if needed.

In many CV tasks, the performance of deep learning models may deviate for people of different genders or races due to bias in training data, leading to unfairness Karkkainen & Joo (2021); Das et al. (2018); Zhang et al. (2018a); Hardt et al. (2016); Zemel et al. (2013). To address bias in AI models, several efforts have been devoted to building fair training datasets (Karkkainen & Joo, 2021; Harvey, 2021; Whitelam et al., 2017). Another line of work aims to improve the algorithms

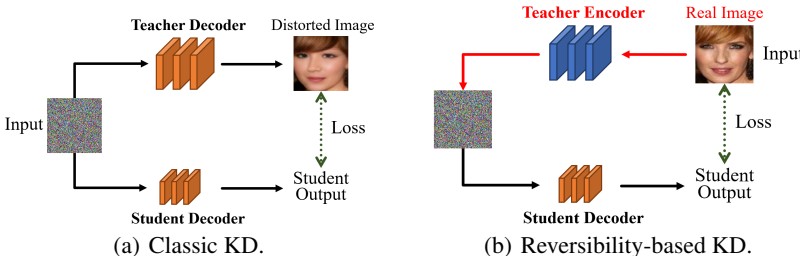

(a) Classic KD.  (b) Reversibility-based KD.

Figure 1: Comparison with the classic KD strategy and reversibility-based KD strategy.

themselves, making AI models more fair through various debiasing techniques during training (Park et al., 2022; Zietlow et al., 2022; Hirota et al., 2022; Das et al., 2018; Zhang et al., 2018a; Hardt et al., 2016). Both the above approaches involve re-training the AI models, which incurs significant costs and is not dependent on the users. In this work, our privacy protection model can process the test dataset to achieve fairness results on already-trained biased AI models. Our goal here for fairness is to let people of different races or genders have similar accuracy when using various AI predictors after we replace the original faces with our generated fake faces. The main contributions of this work are:

1. We present a novel KD method that effectively distills lightweight reversible flow-based generative models. Our approach focuses on mutually transferring knowledge between the encoder and decoder components while maintaining model reversibility.

2. We analyze the vulnerability of the proposed privacy defender, based on the distilled reversible generator, in three different scenarios, under the premise that the intruder obtains different levels of prior information.

3. We implemented a real-time privacy protection system on embedded devices. This system offers efficient privacy preservation while also improving fairness. We deployed and thoroughly evaluated the system on NVIDIA Jetson TX2 devices, demonstrating its effectiveness in real-world scenarios.

## 2 RELATED WORK

**Privacy protection**: In visual privacy, face privacy is a significant part as the face is the most direct identity information. Therein, obfuscation-based methods (McPherson et al., 2016; Yang et al., 2022) are usually adopted for anonymization and synthesis-based methods (Zhu et al., 2020; Ren et al., 2018; Hukkelås et al., 2019; Maximov et al., 2020; Gu et al., 2020; Cao et al., 2021; Zhu et al., 2023) can be used for face anonymization/pseudonymization. This series of synthesis-based methods replace original faces with new faces selected from a pre-described database or fake faces generated based on the original faces through a generative model. In (Zhu et al., 2020), face-swapping technology is used to protect the privacy of patients in medical datasets. It protects privacy by replacing original faces with faces from open-source datasets. In (Ren et al., 2018; Hukkelås et al., 2019; Maximov et al., 2020), GANs are used to generate fake faces while ensuring the reality of the images. However, their methods only consider the anonymization process and cannot recover the original face. (Gu et al., 2020; Cao et al., 2021) combine GANs with passwords to recover the original face when the user inputs the correct password and generate fake faces with different identities when different passwords are used. In (Zhu et al., 2023), the flow-based model is used to anonymize faces by manipulating the latent space to generate different fake faces. Most existing anonymization methods are hard to deploy on devices as they involve complex generative models.

**Knowledge distillation**: KD has been widely applied to various computer vision tasks such as image classification (Romero et al., 2014; Ahn et al., 2019), object detection (Chen et al., 2017; Dai et al., 2021), and semantic segmentation (Liu et al., 2019; Qin et al., 2021). However, KDs for these high-level tasks are often not directly applicable to generative models due to several reasons (Wang et al., 2020; Chen et al., 2020; Li et al., 2022). First, generative models require a more complex and high-dimensional mapping. Second, unlike high-level tasks, generative tasks usually do not have ground-truth (GT) to evaluate outputs. Third, the architectures of generative models are often more sophisticated and have more parameters compared to the models used for high-level tasks. There are

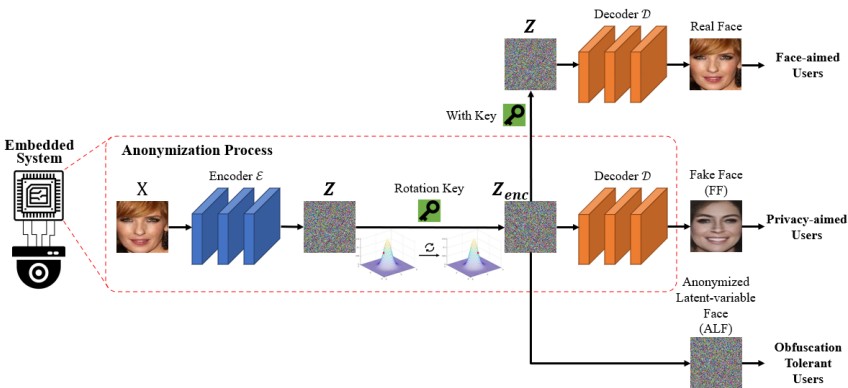

Figure 2: Framework of visual privacy protection.

several works dedicated to the distillation of generative models. (Chen et al., 2020; Li et al., 2020; Jin et al., 2021; Ren et al., 2021; Zhang et al., 2022) proposed a series of distilling methods for GANs. In (Luhman & Luhman, 2021; Salimans & Ho, 2022; Meng et al., 2022), several distillation methods for DMs are proposed to increase their sampling speed. Currently, very little work has been done to distill flow-based models. (Baranchuk et al., 2021) proposed a distillation method for the flow-based models, but they only distilled the decoders.

## 3 BACKGROUND ON THE PRIVACY-PRESERVING STRATEGY

In this section, we first introduce a general framework for visual privacy preservation. We will see how to adopt flow-based models in this framework. At the same time, we will explain why the strong reversibility of the flow model will make it bulky and difficult to embed in local devices.

### 3.1 THE PRIVACY-PRESERVING FRAMEWORK

The proposed anonymization framework for privacy protection is shown in Figure 2. Assume that $\mathbf{X} \sim \mathcal{I}$ is the visual content related to personal privacy in an image, such as a face or license plate, where $\mathcal{I}$ is the image space. Our goal is to hide private information by replacing the original image with a generated fake image. To achieve this goal, we use an encoder $\mathcal{E}$ to map $\mathbf{X}$ to a Gaussian latent space $\mathcal{N}(\mathbf{0}, \mathbf{I})$ by $\mathbf{Z} = \mathcal{E}(\mathbf{X})$, $\mathbf{Z} \sim \mathcal{N}(\mathbf{0}, \mathbf{I})$, where $\mathbf{Z}$ is the latent variable of $\mathbf{X}$. We then use a key to rotate $\mathbf{Z}$ by $\mathbf{Z}_{\mathbf{enc}} = \mathbf{AZ}$, where $\mathbf{Z}_{\mathbf{enc}}$ could be seen as the anonymized feature and $\mathbf{A}$ is an orthogonal matrix that serves as the key. This process can be seen as rotating the encoded feature in the latent Gaussian space and $\mathbf{Z}_{\mathbf{enc}}$ can be sent to obfuscation-tolerant users. Meanwhile, we define a decoder $\mathcal{D}$ such that: $\mathbf{X}' = \mathcal{D}(\mathbf{Z})$, that maps the latent variable $\mathbf{Z}$ back to the image space $\mathcal{I}$. The similarity between the original faces $\mathbf{X}$ and de-anonymized faces $\mathbf{X}'$ depends on the reversibility of the encoder-decoder pair. With reversible $\mathcal{E}$ and $\mathcal{D}$, privacy-aimed users without the key will decode a fake substitute $\mathbf{X}_{\mathbf{fake}} = \mathcal{D}(\mathbf{Z}_{\mathbf{enc}})$, while face-aimed users holding the correct rotation key $\mathbf{A}$, $\mathbf{Z}_{\mathbf{enc}}$ will restore the visual content by $\mathbf{X}' = \mathcal{D}(\mathbf{A}^{-1}\mathbf{Z}_{\mathbf{enc}}) = \mathcal{D}(\mathbf{Z})$. As shown in Figure 2, with different de-anonymization orders, the proposed privacy protection system can support users with different privacy needs. The fake face method (**FF**) uses the fake face as anonymized data for privacy-aimed users, and the anonymized latent-variable face method (**ALF**) uses the latent-variable face as anonymized data for obfuscation-tolerant users.

We can also decrease the variance of $\mathbf{Z}_{\mathbf{enc}}$ to less the variance of the generated fake face $\mathbf{X}'_{\mathbf{fake}} = \mathcal{D}(\alpha\mathbf{Z}_{\mathbf{enc}})$, where $\alpha$ is a control parameter. The variance of the latent Gaussian will affect the diversity of the generated images, that is, the closer $\alpha$ is to 0, the less diverse the faces. This means that the decoder $\mathcal{D}$ will produce a more "average" face than the original faces. In some cases, we might achieve some kind of "fairness" from a visual perspective if we replace the original faces with generated faces that are less diverse. See Appendix A.3.2 for images generated under different $\alpha$.

### 3.2 THE FLOW-BASED PRIVACY-PRESERVING METHOD

In the above privacy-preserving framework, the encoder-decoder pair $\mathcal{E} \sim \mathcal{D}$ could be any model that maps the image space $\mathcal{I}$ to the Gaussian latent space $\mathcal{N}$. A possible choice could be the flow-based model in (Zhu et al., 2023). The flow-based model is bijective, that is, its encoder and decoder

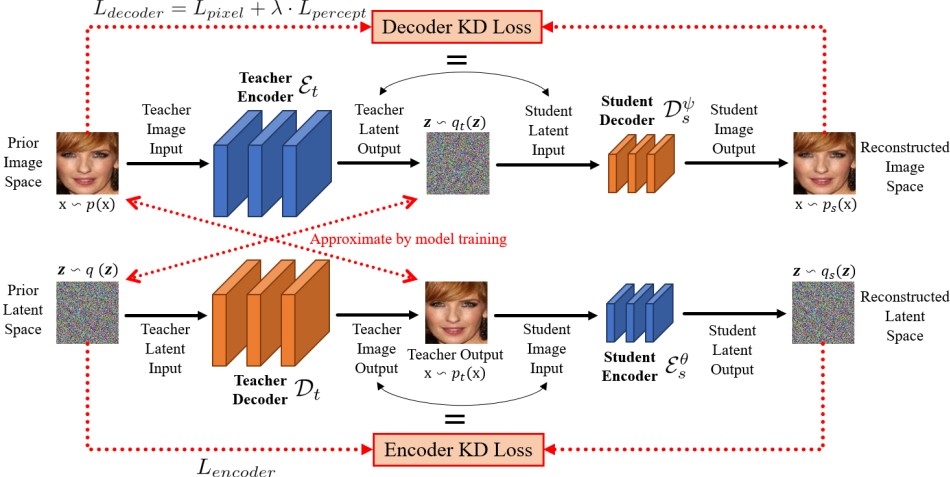

Figure 3: The distillation framework of our method.

are actually the forward process and reverse process of the same model, thereby sharing the same parameters: $\mathcal{E} = \mathcal{D}^{-1} = \mathcal{F}_\theta(\cdot)$, where $\mathcal{F}_\theta(x)$ is a bijection function such that $x = \mathcal{F}_\theta^{-1}(\mathcal{F}_\theta(x))$. We collect a discrete i.i.d. dataset $D = \{x_1, x_2, ..., x_n\} \subset \mathcal{I}$, where $n$ is the number of samples. Let $x_i \sim p(x)$ follow the unknown distribution of the real image data and $p_\theta(x)$ be the distribution of the generated image with model parameter $\theta$, which is also the distribution of the flow decoder output. The goal of training a flow-based model is to learn the parameter $\theta$ such that $p_\theta(x)$ can approximate $p(x)$ well, which is equivalent to minimizing the log-likelihood function: $L(D) = \frac{1}{n}\sum_{i=1}^{n} -\log p_\theta(x_i)$, with $p_\theta(x) = q(z) \cdot \left|\det\left(\frac{\partial x}{\partial z}\right)\right|$, where $z \sim q(z)$ is a prior multivariate Gaussian distribution and $\frac{\partial x}{\partial z}$ is the Jacobian matrix of $\mathcal{F}_\theta^{-1}$.

### 3.3 STRONG REVERSIBILITY AND COMPLEXITY OF FLOW

The flow-based models employ multiple layers of reversible transformations to map the prior data distribution to a simple Gaussian latent distribution, where each layer must satisfy two constraints of reversibility and tractable Jacobian determinants. This makes flow equipped with a large number of parameters, slowing down the inference speed of the model. Therefore, flow-based models are difficult to deploy on devices with low computing resources and limited memory.

## 4 DISTILLATION FOR GENERATIVE MODELS WITH ENCODER AND DECODERS

This section elaborates on the details of the reversibility-based mutually collaborative KD strategy and we will utilize it to compress the generators with an encoder-decoder architecture.

### 4.1 DISTILLATION SCHEME BASED ON ENCODER-DECODER RELATION

Most existing KD methods involve optimizing a loss function defined on the output of the teacher model to transfer knowledge to the student model, as depicted in Figure 1(a). However, for generators that use an encoder-decoder pair, the training process of the teacher encoder can lead to a distorted output space and result in reduced performance for the student encoder if the classic method of distilling the student encoder from the teacher encoder is adopted. In this work, we propose to exploit the encoder-decoder pair's reversibility to allow the student decoder to reconstruct the image space from the input of the teacher encoder by minimizing the distance between the reconstructed and prior image space; as depicted in Figure 1(b). Similarly, we can also distill the student encoder from the teacher decoder by minimizing the distance between the reconstructed latent space and the prior latent space. Our method is then divided into two parts, namely decoder distillation and encoder distillation, and these two parts are mutually collaborative and can be performed in parallel.

**Decoder distillation**: In Figure 3, to distill the teacher decoder, we match the teacher encoder $\mathcal{E}_t$ and the student decoder $\mathcal{D}_s^\psi$ to reconstruct the image space. By sampling an image $\mathbf{x}$ following the prior image distribution $p(\mathbf{x})$, we use $\mathcal{E}_t$ to map $\mathbf{x} \sim p(\mathbf{x})$ to a latent space $q_t(\mathbf{z})$, and then use $\mathcal{D}_s^\psi$ to reconstruct the image from the latent distribution $q_t(\mathbf{z})$. The training process is to minimize the

distance between the original image and the reconstructed image spaces with the pixel-level loss:

$$L_{pixel} = \mathbb{E}_{\mathbf{x} \sim p(\mathbf{x})} \left[ \left\| \mathbf{x} - \mathcal{D}_s^{\psi} \left( \mathcal{E}_t \left( \mathbf{x} \right) \right) \right\|_1 \right]. \tag{1}$$

As using only pixel-level loss will result in blurry generated images (Isola et al., 2017; Chen et al., 2020; Liu et al., 2021), we also incorporate perceptual loss to ensure the student produces perceptually desirable outputs. The perceptual loss is then defined on the learned perceptual image patch similarity (LPIPS) (Zhang et al., 2018b) between the original image and the reconstructed image:

$$L_{percept} = \mathbb{E}_{\mathbf{x} \sim p(\mathbf{x})} \left[ LPIPS \left( \mathbf{x}, \mathcal{D}_s^{\psi} \left( \mathcal{E}_t \left( \mathbf{x} \right) \right) \right) \right]. \tag{2}$$

Therefore, the total training objective of our decoder distillation is $L_{decoder} = L_{pixel} + \lambda \cdot L_{percept}$ with $\lambda$ being the hyperparameter that balances the losses.

**Encoder distillation**: Similarly, to distill the teacher encoder, we match the teacher decoder $\mathcal{D}_t$ and the student encoder $\mathcal{E}_s^{\theta}$ to reconstruct the latent space. Letting $\gamma$ be a hyperparameter for tuning the model, the training loss is defined to minimize the distance between $q(\mathbf{z})$ and $q_s(\mathbf{z})$:

$$L_{encoder} = \mathbb{E}_{\mathbf{z} \sim q(\mathbf{z})} \left[ \left\| \gamma \cdot \mathbf{z} - \mathcal{E}_s^{\theta} \left( \mathcal{D}_t \left( \mathbf{z} \right) \right) \right\|_2^2 \right], \tag{3}$$

where $\mathcal{D}_t$ is used to map $\mathbf{z} \sim q(\mathbf{z})$ to the image space with distribution $p_t(\mathbf{x})$, from which we reconstruct a latent variable with distribution $q_s(\mathbf{z})$ by the student encoder $\mathcal{E}_s^{\theta}$.

It is worth noting that the flow-based model is reversible by nature, which minimizes the distortion between the prior image (latent) space and reconstructed image (latent) space due to the mismatch of the encoder and decoder. Hence, distilling the flow-based model can well show the performance of the proposed mutually collaborative KD strategy. It is expected that the proposed method also has the potential to be extended to other generators without strict reversibility.

## 4.2 THE CNN-BASED STUDENT MODEL

In general, during distillation, one favors homogeneous model architectures for students and teachers. In this work, given that the teacher is flow-based, instead of the flow-based model, we judiciously design a CNN-based student model to achieve a better distillation performance. Due to the page limit, we put the details of the student structure and the training process in Appendix A.1.

## 5 VULNERABILITY ANALYSIS OF THE PRIVACY-PRESERVING METHODS

In this section, we provide a vulnerability analysis of our privacy-preserving model. The setting is to assume that there is an attacker who aims to recover the original image from the anonymized image. Specifically, we consider three types of scenarios:

**Scenario 1:** In this scenario, the attacker only knows the distribution of the original privacy-sensitive image $\mathbf{X}$ and the distribution of anonymized image $\mathbf{X}'_{\mathbf{fake}}$ such that $\mathbf{X}'_{\mathbf{fake}} = \mathcal{D}(\mathbf{A}\mathcal{E}(\mathbf{X})) \triangleq \gamma(\mathbf{X})$. It attempts to crack the anonymization system $\gamma(\cdot)$. Apparently, this is generally impossible as it is well-known that one can not compute joint distribution from marginals without any other prior information (Gelman & Speed, 1993).

**Scenario 2:** In this scenario, the attacker 1) knows the distributions of the original data and the anonymized data; 2) knows the $\mathcal{E} \sim \mathcal{D}$ pair of the flow-based model; 3) does not possess anonymization key $\mathbf{A}$. Apparently, in this case, if the attacker can crack the key $\mathbf{A}$, it can immediately recover the original image through the decoder $\mathcal{D}$. The difficulty in tackling this problem has been studied in (Wu et al., 2020) and we summarize it as Lemma 1.

**Lemma 1 (Wu et al., 2020; Wainwright, 2019; Hoeffding & Wolfowitz, 1958).** *Let the standard Gaussian distribution be $H_0$ and the distribution of the anonymized latent variables be $H_1$. The likelihood of the attacker recovering the key $\mathbf{A}$ does not exceed $\delta \left( H_0^n, H_1^n \right) + \theta$, where $\delta \left( H_0^n, H_1^n \right)$ is the total variation distance (see definitions in Appendix A.2.1) between $H_0^n$ and $H_1^n$, $n$ is the number of samples, and $\theta$ is the tolerance during the recovery of key $\mathbf{A}$.*

Lemma 1 tells that the probability that the attacker recovers key $A$ depends on the distance between the distribution $H_1^n$ of the anonymized latent variables and the standard Gaussian distribution $H_0$. When $H_1^n$ is also a standard Gaussian distribution, this upper limit is the smallest, that is, it is most difficult for the attacker to recover the key $A$. More details can be found in Appendix A.2.1

Table 1: Distillation result on face datasets and CIFAR10. On these two datasets, we can compress the model to $14.7M$ and $16.1M$ parameters respectively and maintain satisfactory performance.

| Scheme | CelebA & FFHQ | | | | | CIFAR10 | | | | |
|---|---|---|---|---|---|---|---|---|---|---|
| | Param (M) | FLOPs (B) | FID↓ | LPIPS↓ | PSNR↑ | Param (M) | FLOPs (B) | FID↓ | LPIPS↓ | PSNR↑ |
| Teacher | 37.0 | 16.52 | 35.79 | **0.000** | ∞ | 37.0 | 16.52 | 77.07 | **0.000** | ∞ |
| MSE | | | 49.26 | 0.206 | 19.90 | | | 104.04 | 0.257 | 17.97 |
| LPIPS | | | 47.65 | 0.145 | 18.06 | | | 93.15 | 0.257 | 17.22 |
| Baranchuk | 18.1 | 2.71 | 45.85 | 0.157 | 19.69 | 18.1 | 2.71 | 92.04 | 0.264 | 17.68 |
| OMGD | | | 50.59 | 0.176 | 19.49 | | | 107.59 | 0.247 | 17.48 |
| WKD | | | 49.38 | 0.201 | **20.03** | | | 103.04 | 0.230 | 17.96 |
| | 18.1 | 2.71 | **22.47** | **0.136** | **20.03** | 18.1 | 2.71 | **69.55** | **0.213** | **18.10** |
| **Ours** | 16.1 | **2.38** | 27.82 | 0.146 | 19.96 | **16.1** | **2.38** | 72.27 | 0.226 | 17.76 |
| | **14.7** | 2.68 | 30.53 | 0.164 | 19.31 | - | - | - | - | - |

**Scenario 3:** In this case, the attacker can access our terminal privacy-preserving device to obtain a batch of paired original and anonymized data. This is the strongest attacking case as the attacker can train a neural network in a supervised manner to learn the proposed system (Salem et al., 2020; Yang et al., 2019; Cunningham et al., 2008). We simulated this scenario in our experiments in Section 6.C.

## 6  EXPERIMENTS

In this section, we first evaluate the effectiveness of the proposed distillation method by comparing it with several existing distillation methods designed for generative models and then test the models' privacy-preserving ability, reversibility, data usability, as well as the ability of fairness enhancement.

**Models and datasets**: Our experiments have been conducted on the CelebA (Liu et al., 2015), FFHQ (Karras et al., 2019), CIFAR10 (Krizhevsky et al., 2009), LFW (Huang & Learned-Miller, 2014), UTK-face (Zhang et al., 2017) and HMDB51 (Kuehne et al., 2011). For CelebA and FFHQ, we mixed them, detected the faces using YOLO5Face (Qi et al., 2023), and cropped them to achieve better generalization and stronger face-generation performance. We used $90\%$ of the resulting data as the training set and $10\%$ as the test set. The teacher model adopted is Glow (Kingma & Dhariwal, 2018) and it is trained on CelebA & FFHQ, which has been also used in (Zhu et al., 2023). The student model is obtained from the teacher by the proposed KD method according to Appendix A.1.

### A. DISTILLATION RESULTS

**Evaluation metrics**: We evaluate the proposed distillation method from three aspects: 1) **Image generation quality**: lower Fréchet Inception Distance (**FID**) (Heusel et al., 2017) indicates higher image quality and diversity; 2) **Model reversibility**: **LPIPS** (Zhang et al., 2018b) and **PSNR** are metrics for measuring image similarity, higher PSNR, and lower LPIPS indicating stronger reversibility; 3) **Model size**: parameter count (**Param**) and floating point of operations (**FLOPs**).

**Baselines**: The baseline KDs include: **(1) MSE**: directly using mean squared error (MSE) to align the outputs of the teacher and student models, which is the simplest method to distill generative model as shown in (Aguinaldo et al., 2019; Luhman & Luhman, 2021). **(2) LPIPS**: using LPIPS (Zhang et al., 2018b) to align the outputs of the teacher and student models. **(3) Baranchuk**: the only method we found for distilling conditional flow-based model (Baranchuk et al., 2021). **(4) OMGD**: a distillation method for GAN (Ren et al., 2021). **(5) WKD**: a response-based KD for GAN (Zhang et al., 2022), but can also be used for flow-based models.

**Comparisons**: The KD experimental results are as shown in Table 1. Regarding the quality of the generated images, our method achieves lower FID scores than counterpart distillation methods. Moreover, the counterpart methods harm the performance of the student's generated images (with higher FID than the teacher), while our method enables the student to have better image generation ability than the teacher (with lower FID than the teacher) although the size of the student is much smaller. In terms of model reversibility, our method achieves lower LPIPS and higher PSNR compared to other methods. This indicates that our method can result in a student model with better reversibility. The cost of the KD process is provided in Appendix A.4.

### B. STUDENT PERFORMANCE FOR PRIVACY-PRESERVING

We organize our experiments by testing the effectiveness of privacy-preserving ability, the utility of anonymized data, and fairness. We verify the two privacy-preserving methods: the fake face method

(**FF**) (using the fake face in Figure 2 as anonymized data) and the anonymized latent-variable face method (**ALF**) (using the anonymized latent-variable face in Figure 2 as anonymized data). The difference between ALF and FF is whether we rotate the latent-variable face or not.

**Effectiveness of face privacy-preserving**: In the field of computer vision, two indicators are usually adopted to measure whether face privacy is protected well. One is how well the original human face is seen by human visions (HV) (Zhao et al., 2023), and the other is whether information closely related to the original faces, such as identity, race, and gender, will be recognized by computer vision (CV) machines (Maximov et al., 2020; Zhu et al., 2023; Gu et al., 2020; Zhao et al., 2023). From an HV perspective, we show the similarity between original and anonymized images, compared with some traditional methods (e.g., blacked out, pixelation, random fake face), and differential privacy with given $\epsilon$ (Dwork, 2006). Higher MSE and LPIPS indicate lower image similarity and better privacy protection. We randomly sample 90k original images from the CelebA and FFHQ datasets and anonymize them. Table 2 shows that, compared with the traditional and commonly used anonymization method, the image anonymized by the ALF method of student and teacher leaks the least private information. The similarity between the anonymized face obtained by the student FF method and the original face is also significantly worse, which shows that the anonymized face by the student model protects private information. From a CV perspective, we evaluate the effectiveness of privacy protection against four commonly employed third-party "black-box" FR models, namely FaceNet (Schroff et al., 2015), SphereFace (Liu et al., 2017), CosFace (Wang et al., 2018a), and ArcFace (Deng et al., 2019). Following the setting in (Yang et al., 2021; Zhu et al., 2023), we randomly select $500$ faces with different identities from the LFW dataset as probe images, while using the remaining images as the gallery. The aforementioned FR models are utilized to perform identity recognition on the probe images. For recognition, if the correct identity is included in the top $N$ identities ranked by similarity, it is recorded as correct. The results are shown in Table 3. All four FR models achieve an accuracy of over $89.4\%$ in identifying the original faces. However, the accuracy of identifying anonymized faces is almost $0\%$ for all FR models, regardless of whether the teacher or student model is used for privacy preservation. This indicates that privacy protection can be achieved by utilizing the student in our system.

We also show the anonymization and de-anonymization faces of the student and teacher in Appendix A.2.2 and the data usability for downstream high-level tasks in Appendix A.2.3. The results show that using our student model does not affect the effectiveness of downstream high-level tasks.

Table 2: The similarity between the original image and the anonymized image.

| | Blacked out | Pixelation | Random face | DP ($\epsilon = 5$) | DP ($\epsilon = 2$) | Tea. ALF | Tea. FF | Stu. ALF | Stu. FF |
|---|---|---|---|---|---|---|---|---|---|
| MSE↑ | 0.302 | 0.034 | 0.084 | 0.080 | 0.500 | **1.316** | 0.129 | **0.557** | 0.126 |
| LPIPS↑ | 0.794 | 0.665 | 0.438 | 0.702 | 0.819 | **0.860** | 0.738 | **0.837** | 0.727 |

Table 3: Top-$N$ accuracy of face recognition.

| | | FaceNet Top-1 | FaceNet Top-5 | CosFace Top-1 | CosFace Top-5 | ArcFace Top-1 | ArcFace Top-5 | SphereFace Top-1 | SphereFace Top-5 |
|---|---|---|---|---|---|---|---|---|---|
| Original | | 93.4% | 97.8% | 95.4% | 97.8% | 96.0% | 98.2% | 89.4% | 93.4% |
| Tea. | ALF | 0% | 0% | 0% | 0.6% | 0% | 0.2% | 0% | 0% |
| | FF | 0% | 0% | 0% | 0.6% | 0% | 0% | 0% | 0.2% |
| Stu. | ALF | 0% | 0.4% | 0% | 0% | 0% | 0% | 0.2% | 0.6% |
| | FF | 0% | 0% | 0% | 0.2% | 0% | 0.4% | 0% | 0% |

**Fairness enhancement**: In this part, we will show that if we process the test dataset with our privacy-preserving system, general AI predictors will provide more unbiased results on different groups. To proceed, we first use YOLO5Face (Qi et al., 2023) to perform face detection on UTK-face and consider the results as ground truth (GT). Then, we utilize the face detection model in the OpenCV library (Bradski & Kaehler, 2008) to perform face detection on the original images and anonymized images and calculate the intersection over union (IoU) between the results and the GT. For gender and race attributes, we counted the mean IoU of the OpenCV detector in each group and calculated the difference between each group as metrics of fairness, similar to the previous work (Karkkainen & Joo, 2021). See Appendix A.3.1 for details of the experiment. The fairness results in Table 4 show that the OpenCV detector is biased on the original images, and after anonymization by either the teacher or the student models, this bias is alleviated. This result indicates that using the student model to anonymize face images does enhance fairness.

Table 4: Face detection results for different groups and fairness among different groups. "Ori.", "Tea." and "Stu." denote the results of the original images, the images anonymized by the teacher, and the images anonymized by the student, respectively. Regarding the results for "*Gender*", we don't report $D_{mean}$ and STD since there are only two groups.

| | Gender | | | | Race | | | | | | | | |
| | mIoU | | $D_{max}\downarrow$ | $\epsilon\downarrow$ | mIoU | | | | | $D_{max}\downarrow$ | $D_{mean}\downarrow$ | $\epsilon\downarrow$ | STD$\downarrow$ |
| | Male | Female | | | White | Black | Asian | Indian | Others | | | | |
| Ori. | 0.665 | 0.613 | 0.052 | 0.081 | 0.623 | 0.596 | 0.643 | 0.669 | 0.617 | 0.073 | 0.034 | 0.115 | 0.028 |
| Tea. | 0.686 | 0.665 | **0.021** | **0.031** | 0.666 | 0.659 | 0.668 | 0.690 | 0.643 | **0.047** | **0.020** | **0.070** | **0.017** |
| Stu. | 0.695 | 0.670 | **0.025** | **0.037** | 0.674 | 0.673 | 0.678 | 0.696 | 0.673 | **0.022** | **0.010** | **0.033** | **0.010** |

Table 5: Similarity between various faces and the original face.

| | MSE↑ | LPIPS↑ |
| --- | --- | --- |
| Random face | 0.084 | 0.438 |
| Fake face | 0.126 | 0.727 |
| Attacked face | 0.055 | 0.710 |

Table 6: Inference time and energy consumption on embedded devices.

| Method | | Time (ms) | Energy (J) | | |
| --- | --- | --- | --- | --- | --- |
| | | | GPU | CPU | Board |
| Tea. | ALF | 57.88 | 0.416 | 0.009 | 0.770 |
| Stu. | ALF | 10.46 | 0.075 | 0.002 | 0.143 |
| | FF | 12.67 | 0.088 | 0.002 | 0.172 |

## C. VULNERABILITY EXPERIMENT

We simulate the vulnerability experiment for Scenario 3 in Section 5. The attacker tries to reverse the proposed anonymization model by using supervised learning. That is, the input of the learning network is the anonymized image, and the learning target of the output is the corresponding original image. We may choose any suitable neural network to learn the anonymization model. Herein, we choose to adopt the same architecture as the anonymization model to achieve a better attack result. The training dataset is 100k images randomly selected from the face datasets CelebA and FFHQ, and then anonymized with our student model (by encoding, rotating and decoding process) to obtain the anonymized images. Herein, "Random face", "Fake face", and "Attacked face" respectively represent faces randomly selected from the dataset, fake faces anonymized by the student model, and faces obtained by attacking the student model. Table 5 shows that even if an attacker can obtain a large number of original-anonymized image pairs, it is still difficult to restore the original image.

## D. EMBEDDED DEVICES DEPLOYMENT

To demonstrate that the compressed model can run on an embedded system and achieve real-time performance, we deploy our system on the NVIDIA Jetson TX2, which is an embedded system module designed and produced by NVIDIA. It utilizes NVIDIA's Tegra X2 processor, which has 256 CUDA cores and 8 ARM CPU cores and comes with 8GB LPDDR4 memory. To reduce memory usage and optimize inference performance, we deploy our models with TensorRT (https://developer.nvidia.com/tensorrt) a high-performance deep learning inference optimizer and runtime library. We test the inference time and required energy for ALF using the teacher model and student model separately. Moreover, we deploy FF on embedded devices using our student model, which is not achieved in FFEM. We anonymized 10k images on the embedded system and obtained average inference time and energy consumption. The results are shown in Table 6. For ALF, the inference speed of the student is 10.46 ms, which is 5.53 times faster than that of the teacher, achieving much faster real-time inference speed. And the student only needs 0.143J to anonymize one image, which is much lower than the teacher. For FF, students can anonymize an image using only 12.67ms and 0.172J of energy, enabling real-time inference on embedded devices.

## 7 CONCLUSIONS AND LIMITATIONS

**Conclusions**: We propose a novel reversibility-based knowledge distillation method for flow-based generative models. This method leverages the relation between the encoder and decoder to effectively preserve the reversibility. Based on the lightweight and reversible generative model, we embed our privacy protection and fairness enhancement system on a real NVIDIA Jetson TX2 device.

**Limitations**: Although using our method to obtain the student model results in stronger reversibility compared to other methods, the reversibility still decreases compared to the teacher model, which results in some differences between the recovered image and the original image. We believe that this is a worthwhile trade-off between reversibility and privacy-preserving capability.

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
