# A APPENDIX

## A.1 DETAIL OF MUTUALLY COLLABORATIVE DISTILLATION

### A.1.1 ARCHITECTURES OF CNN-BASED STUDENT MODELS

In this work, we judiciously design a CNN-based student model to achieve a better distillation performance for two reasons: First, the flow-based model has reversibility and tractable Jacobian determinant constraints (Dinh et al., 2014; 2016), both of which should be taken into account in the loss function during the flow training process. This will affect the optimization of the distillation loss and ultimately lead to poor performance of the student model. Second, reversibility and tractable Jacobian determinant in each layer reduce the mapping capacity of each layer in the network (Dinh et al., 2014; 2016). Thus, to accomplish a learning task, more layers are required, increasing the model's size. Compared with the flow-based students, each layer of the CNN-based student model has a stronger mapping ability, leading to a smaller size encoder-decoder and faster inference ability. A similar CNN-based student model first appeared in (Baranchuk et al., 2021) for compressing flow-based models for super-resolution tasks.

In particular, the CNN-based student models are designed following the architecture of the flow-based model, while we replace the invertible $1 \times 1$ convolution layers of the flow-based model with the standard $1 \times 1$ and $3 \times 3$ convolution layers and replace the coupling layers of the flow-based model with the residual dense blocks (RDB) in (Zhang et al., 2018c). Other structures of the flow-based model such as the squeeze layer and the actnorm layer remain unchanged. The specific architecture is shown in Figure 4, where subfigure (a) is the encoder of the flow-based model and subfigure (b) is the corresponding CNN-based student encoder with the replacement of convolutional layers, the architectures of the student encoder and decoder are fully mirrored, and "$K$" and "$L$" are two hyperparameters that control the size of the model.

The main differences between the invertible $1 \times 1$ convolution layers and the standard convolution layer are: for the former structure 1) the input and output dimensions must be the same, 2) the convolution kernel size must be $1 \times 1$, and 3) the convolution kernel parameter matrix must be reversible, while the latter structure does not have these restrictions. These three limitations in the former structure actually limit the dimension of feature maps and inhibit the ability of the network to extract features. Therefore, we replace them with CNN-based structures in the student model to achieve better compression capabilities.

Similarly, the reason why we use RDB to replace the coupling layers is that RDB has stronger feature extraction capabilities and can make full use of features at various levels (Zhang et al., 2018c). Actually, RDB has been widely used in the field of computer vision, especially in super-resolution tasks (Zhang et al., 2018c; Wang et al., 2018c) and image restoration tasks (Zhang et al., 2020; Kim et al., 2019), and has achieved excellent performance.

Note that, once we change the invertible $1 \times 1$ convolution layers and the coupling layers in the original flow-based model, we have destroyed the fully-reversibility and tractable Jacobian determinant constraints of the flow-based model. However, our experiments show that the CNN-based student model can still have decent privacy-preserving ability and kept much of the reversibility.

### A.1.2 TRAINING DETAILS

When training the teacher model, we use the Adam optimizer, where $\beta_1$ and $\beta_2$ are set to $0.9$ and $0.999$, and the learning rate linearly increases from $0$ to $0.0001$ over $50k$ steps. Batch size, $K$, and $L$ are set to $64$, $16$, and $4$. The teacher model is trained for $500$ epochs.

When training the student models by KD, we use the same optimizer as used during the training of the teacher model, and the learning rate linearly increases from $0$ to $0.0001$ over $20k$ steps. Batch size, $\lambda$, and $\gamma$ is set to $128$, $10^4$, and $10^4$ respectively. For the student encoder, $K$ and $L$ are set to $6$ and $3$. For the student decoder, $K$ and $L$ are set to $(2, 3)$, $(1, 3)$, and $(3, 2)$ respectively for different size model. All student models are trained for $800$ epochs.

### A.1.3 ALGORITHM

The pipeline of our method is sketched in Algorithm 1.

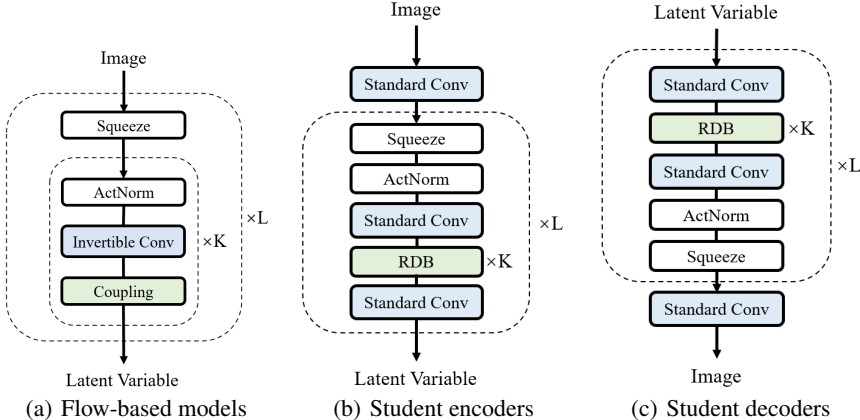

Figure 4: Architectures of flow-based models and students. "Squeeze", "ActNorm", "Invertible Conv", and "Coupling" are the squeeze layer, actnorm layer, invertible $1 \times 1$ convolution layer, and coupling layer, which are the standard modules in flow-based models (Kingma & Dhariwal, 2018). "Conv" is the standard convolution layer.

---

**Algorithm 1** Reversibility-based knowledge distillation

---

**Require:** prior image distribution $p(\mathbf{x})$, prior latent distribution $q(\mathbf{z})$, teacher encoder $\mathcal{E}_t$, teacher decoder $\mathcal{D}_t$

**Ensure:** student encoder $\mathcal{E}_s^\theta$ with parameters $\theta$, student decoder $\mathcal{D}_s^\psi$ with parameters $\psi$

1: **while** not converged **do**
2:     $\mathbf{x} \sim p(\mathbf{x})$                                                    ▷ Sample image
3:     $\mathbf{x}' = \mathcal{D}_s^\psi \left(\mathcal{E}_t\left(\mathbf{x}\right)\right)$                                 ▷ Reconstruct image
4:     $L_{pixel} = \left\| \mathbf{x} - \mathbf{x}' \right\|_1$
5:     $L_{percept} = LPIPS\left(\mathbf{x}, \mathbf{x}'\right)$
6:     $L_{decoder} = L_{pixel} + \lambda \cdot L_{percept}$
7:     $\psi \leftarrow \psi - \eta \nabla_\psi L_{decoder}$
8: **end while**
9: **while** not converged **do**
10:     $\mathbf{z} \sim q(\mathbf{z})$                                            ▷ Sample latent variable
11:     $\mathbf{z}' = \mathcal{E}_s^\theta \left(\mathcal{D}_t\left(\mathbf{z}\right)\right)$                        ▷ Reconstruct latent variable
12:     $L_{encoder} = \left\| \gamma \cdot \mathbf{z} - \mathbf{z}' \right\|_2^2$
13:     $\theta \leftarrow \theta - \eta \nabla_\theta L_{encoder}$
14: **end while**

---

## A.2 SUPPLEMENTARY EXPERIMENTS FOR PRIVACY

### A.2.1 DEFINITIONS FOR LEMMA 1 (WU ET AL., 2020)

**Definition 1 (total variation distance).** *For two distributions $P$ and $Q$ defined on domain $\mathcal{D}$, the total variation distance is defined as*

$$\delta(P, Q) \triangleq \sup_{A \in \mathcal{D}} |P(A) - Q(A)| \tag{4}$$

**Definition 2 (normalized volume).** *Let $U$ be the set of all orthogonal matrices with size $m \times m$, and let $T$ be a subset of $U$. Define the normalized volume of $T$ be*

$$v(T) \triangleq \mathbb{P}[A \in T], \tag{5}$$

*where $A$ is sampled uniformly from all $m \times m$ orthogonal matrices.*

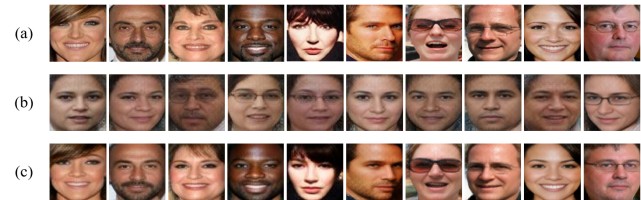

Figure 5: Face privacy-preserving ability using student model. (a) Original face randomly selected from CelebA and FFHQ. (b) Anonymized fake face. (c) De-anonymized face.

Table 7: Top-$N$ accuracy of video action recognition.

|  |  | Split 1 | | Split 2 | | Split3 | |
|---|---|---|---|---|---|---|---|
|  |  | Top-1 | Top-5 | Top-1 | Top-5 | Top-1 | Top-5 |
| Original | | 65.56% | 90.46% | 91.18% | 96.60% | 90.98% | 96.86% |
| Tea. | ALF | 65.56% | 90.39% | 91.18% | 96.81% | 90.94% | 96.88% |
| | FF | 65.17% | 89.87% | 90.36% | 96.47% | 90.21% | 96.74% |
| Stu. | ALF | 65.44% | 90.49% | 91.04% | 95.97% | 90.39% | 96.74% |
| | FF | 65.19% | 89.67% | 90.33% | 96.51% | 91.06% | 96.86% |

**Definition 3 ($(\theta, A)$-ball).** *Let $U$ be the set of all orthogonal matrices with size $m \times m$. Given an orthogonal matrix $A \in \mathbb{R}^{m \times m}$, $\rho > 0$, the ball centered at $A$ with radius $\rho$ is defined as*

$$\mathcal{B}_\rho(A) \triangleq \{M | \|M - A\|_F \leq \rho, M \in U\}. \tag{6}$$

*Given any $\theta \in (0, 1]$, let $\rho^* \triangleq \min\{v(\mathcal{B}_\rho(A)) \geq \theta\}$. We define $(\theta, A)$-ball as $\mathcal{B}_{\rho^*}(A)$.*

**Definition 4 (successful recovery with tolerance $\theta$).** *A recovery of matrix $A$ by an adversary is a successful recovery with tolerance $\theta$ if the output $A'$ of the adversary is within the $(\theta, A)$-ball.*

In other words, if the output $A'$ is close to $A$ in terms of Frobenius norm, we say the recovery is successful. The $\theta$ tolerance term is controlled by the data provider.

### A.2.2 THE ANONYMIZATION AND DE-ANONYMIZATION FACE

The anonymization and de-anonymization face images of the student model are shown in Figure 5. We can see that even although Tables 2 and 3 show that it could be difficult to identify the original identity from the anonymized images from HV and CV perspectives, the de-anonymized images are very close to the original images.

### A.2.3 EFFECT ON DOWNSTREAM HIGH-LEVEL TASKS

To verify data usability for downstream high-level tasks, we perform a video action recognition task. The video dataset used is HMDB51, which includes videos of 51 different action classes. The action recognition model adopts 3D-ResNet (Hara et al., 2018). We fine-tune the action recognition model on the HMDB51 dataset using the pre-trained model provided in (Hara et al., 2018). Top-$N$ is utilized as the evaluation metric for action recognition. The results are presented in Table 7. Compared to the accuracy before anonymization, the accuracy using teacher anonymization and student anonymization dropped slightly and they are nearly identical. This shows that using our student model in our system does not affect the effectiveness of downstream high-level tasks.

### A.3 SUPPLEMENTARY EXPERIMENTS FOR FAIRNESS

### A.3.1 FAIRNESS METRICS

The metric IoU refers to the ratio of the intersection and union of the detection bounding box and the ground truth bounding box, which is a commonly used metric in object detection. Higher IoU means better detection performance. Let $A = \{a_1, a_2, ...\}$ indicate a set of different groups about

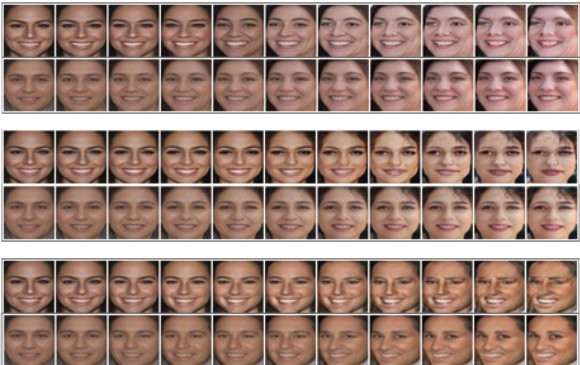

Figure 6: Images generated by the model at different temperatures. Even-numbered rows and odd-numbered rows are images generated by the teacher model and student model respectively. From left to right, $\alpha$ increases from 0 to 1.

a sensitive attribute. $mIoU_{a_i}$ represents the mean IoU of the group whose sensitive attribute is $a_i$. The fairness metrics we use can be formulated as (Karkkainen & Joo, 2021):

$$D_{max} = \max_{\forall a_i, a_j \in A} (mIoU_{a_i} - mIoU_{a_j}), \tag{7}$$

$$D_{mean} = \frac{1}{n} \sum_{a_i \neq a_j} |mIoU_{a_i} - mIoU_{a_j}|, \tag{8}$$

$$\epsilon = \max_{\forall a_i, a_j \in A} \left( \log \frac{mIoU_{a_i}}{mIoU_{a_j}} \right). \tag{9}$$

Moreover, "STD" denotes the standard deviation of results on different groups. The above metrics are defined to measure the degree of difference between groups. Lower levels of these metrics indicate lower bias, i.e. higher fairness.

For the sensitive attribute gender, we divide the UTKFace dataset into two groups: male and female. For the sensitive attribute race, we divide the UTKFace into four groups: White, Black, Asian, and Indian. We then calculate the mIoU of each group and use the difference metrics on IoUs to measure fairness among groups.

### A.3.2 FAIRNESS ENHANCEMENT

We can decrease the variance of $\mathbf{Z_{enc}}$ to less the variance of the generated fake face $\mathbf{X'_{fake}} = \mathcal{D}(\alpha \mathbf{Z_{enc}})$, where $\alpha$ is a control hyperparameter. It is easy to see that the variance of the latent Gaussian will affect the diversity of the generated images, that is, the closer $\alpha$ is to 0, the less diverse the faces. This means that the decoder $\mathcal{D}$ will produce a more "average" face than the original faces.

In some cases, we might achieve some kind of "fairness" from a visual perspective if we replace the original faces with generated faces that are less diverse. This can alleviate the bias of AI models against people with different attributes. As shown in Figure 6, as $\alpha$ increases from 0 to 1 (from left to right), the diversity of images gradually increases.

Table 8: Time required for various distillation method.

|      | MSE   | LPIPS | Baranchuk | OMGD  | WKD   | Ours  |
|------|-------|-------|-----------|-------|-------|-------|
| Time | 2d13h | 2d19h | 2d19h     | 2d14h | 2d14h | 2d17h |

### A.4 COST OF DISTILLATION

In terms of distillation time, we recorded the time required for various distillation methods mentioned in Table 1. Both the teacher model and the student model adopt the exact same structure and the same hyperparameters. All methods use the same hardware environment (NVIDIA 3090) and

the same training hyperparameters (batch size is 128). The results are shown in Table 8. There is not much difference in the time required for various distillation methods.