# OpenReview forum: "Defender of privacy and fairness: tiny but reversible generative model via mutually collaborative knowledge distillation"
_ICLR.cc/2024/Conference — Submitted to ICLR 2024_

### Official Review · Reviewer_u6s5 · 2023-10-18

**Soundness:** 3 good
**Presentation:** 2 fair
**Contribution:** 3 good
**Rating:** 5
**Confidence:** 3

**Summary:**

This paper proposes a mutually collaborative knowledge distillation method to train a tiny and reversible generative model. The resulting student models can be embedded in devices to anonymize private data and improve security. Experiments on face images show that the proposed system protects them through anonymization and also improves fairness without impacting downstream tasks.

**Strengths:**

* Synthesis-based privacy protection is important, and the proposed work is designed to work in embedded devices and is thus practical.
* The mutually collaborative KD method for encoder-decoder pairs looks novel.
* Experiments show that the proposed method has excellent privacy-preservation, reversibility, data usability, and fairness.

**Weaknesses:**

* The fairness protection seems to be more of a side effect instead of a key improvement that is only possible with the proposed method. The proposed architecture does not consider fairness at all and seem to be focused on privacy preservation only. The paper assumes a vague notion of fairness (what does "we might achive some kind of "fairness" from a visual perspective" mean?) instead of the typical fairness notions like group fairness, individual fairness, and causal fairness. The experiments are a bit underwhelming as they do not show how the proposed method improves fairness compared to other generative models.

* More critically, the authors seem to assume that making data private generally improves fairness, which is not necessarily true. An obfuscation can actually be discriminatory against minority groups and actually worsen fairness. This possible tradeoff is not discussed, and there is no convincing argument that there is no such tradeoff either. Hence the proposed method may in fact not be defending fairness as claimed in the title.

* The presentation is sometimes verbose and can be improved. For example, the second to last paragraph on Page 2 is difficult to understand with too many new terms appearing without definitions. As another example, the second paragraph on Page 8 is very long and talks about multiple things.

**Questions:**

Please address the two points about fairness above.

---

### Official Review · Reviewer_yB4B · 2023-10-31

**Soundness:** 2 fair
**Presentation:** 3 good
**Contribution:** 2 fair
**Rating:** 3
**Confidence:** 4

**Summary:**

This work proposed a novel reversibility-based knowledge distillation method for flow-based generative models. This method leverages the relation between the encoder and decoder to effectively preserve the reversibility. Based on the lightweight and reversible generative model, their privacy protection and fairness enhancement system can be embeded on a real NVIDIA Jetson TX2 device. Experiments demonstrate that their system can anonymize face images and also improve fairness while minimizing the impact on downstream tasks.

**Strengths:**

1. The proposed privacy protection system can support users with different privacy needs.
2. The results seem promising.
3. The paper is generally well motivated and written.

**Weaknesses:**

1. The proposed anonymization framework for privacy protection in Figure 2 (Sec. 3.1) seems very naive, no much novelty. Sec. 3.2 and 3.3 seem new, but too short.
2. INTRODUCTION is really too long, ~2 pages, can be trimmed down.
3. this paper covers both privacy and fairness, these 2 topics are both big topics and in many cases will conflict with each other (a trade-off problem). Moreover, fairness has many different concepts, seems this paper did not clearly point out which kind of fairness it aims to tackle. I would recommend mainly focus on 1 topic and make it more solid in a single paper. Also how much privacy can be protected is not well quantified in this work.
4. In baseline KDs, seems MSE and LPIPS are both metrics, rather than KD methods.
5. Lack of detailed info for human visions (HV) assessment, normally people need to ask real human beings from diverse background to help evaluate the privacy the images.
6. Beyond flow-based generative models, I'm wondering whether the proposed reversibility-based knowledge distillation method still works?
7. The reversibility will result in some differences between the recovered image and the original image, it would help a lot to quantify this difference.

**Questions:**

see weakness

---

### Official Review · Reviewer_gq9w · 2023-10-31

**Soundness:** 2 fair
**Presentation:** 2 fair
**Contribution:** 1 poor
**Rating:** 3
**Confidence:** 4

**Summary:**

Motivated by the emerging AI regulations, the paper proposes a knowledge distillation method to distill reversible generative models. It uses a fixed teacher encoder (decoder) to train a lightweight student decoder (encoder). With the models, one can replace privacy-sensitive human faces with synthesized faces and recover the images with a reserved private key. The experiments show that models distilled by the proposed method are able to run on an embedding device in real time.

**Strengths:**

1. How to run large-scale generative models on edge devices is an important and practical problem.

2. The authors conduct various experiments showing that the proposed method can run on edge devices and deliver good numbers on many metrics.

**Weaknesses:**

1. The novelty and contributions are limited. The main contribution seems to be the framework of knowledge distillation, while such a scheme has been studied in prior work [1]. The difference is unclear, at least not specified in the paper.

2. The problem setting seems not convincing. After reading the paper, I still don't get the motivation and in what situation the users will need to recover the faces. I suppose a system that sanitizes information according to the different levels of privacy requirements should be enough, for example, blacking out or performing face swapping for the most sensitive information. Why and when do we need to recover it? Without strong motivations, I doubt the mechanism just offers a new possibility for privacy attacks, though the authors have provided an initiative discussion about the potential risks in Sec. 5.

3. There is no qualitative result in the main paper, making it difficult to evaluate the effectiveness. Though the authors provide visualization in the appendix, the synthetic images look totally different from the original, making me confused again about the motivation. If it aims to execute the original vision tasks, such faces may not be feasible to substitute for the original information. One should consider preserving task-specific information such as [2]. If that's not the goal, why don't we simply use face-swap techniques?

[1] Huan Wang, Yijun Li, Yuehai Wang, Haoji Hu, and Ming-Hsuan Yang. Collaborative distillation for ultra-resolution universal style transfer. In Proceedings of the IEEE/CVF conference on computer vision and pattern recognition, pp. 1860–1869, 2020.

[2] Wang, Hui-Po, Tribhuvanesh Orekondy, and Mario Fritz. "Infoscrub: Towards attribute privacy by targeted obfuscation." Proceedings of the IEEE/CVF Conference on Computer Vision and Pattern Recognition. 2021.

**Questions:**

1. The privacy protection system seems to play a critical role in the paper. I wonder how the system executes. Does one have first to detect the faces and replace the faces with synthetic ones? Are there any such practical experiments in the paper, or do the authors only consider pure face datasets?

2. Is there any practical use case where users need to recover the faces afterward?

---

### Meta-Review · Area_Chair_Vw3c · 2023-12-04

**Metareview:**

The reviewers unanimously agreed that the paper stands below the acceptance threshold, and the authors chose not to respond to reviews and update their manuscript. I encourage the authors to revise their manuscript to reflect the reviews.

**Justification For Why Not Higher Score:**

The reviewers were unanimous in rejecting the paper and the authors did not submit a rebuttal.

**Justification For Why Not Lower Score:**

N/A

---

### Decision · Program_Chairs · 2024-01-16

Reject